# Radioactive Iodine Treatment and the Risk of Long-Term Cardiovascular Morbidity and Mortality in Thyroid Cancer Patients: A Nationwide Cohort Study

**DOI:** 10.3390/jcm10174032

**Published:** 2021-09-06

**Authors:** Chun-Hao Kao, Chi-Hsiang Chung, Wu-Chien Chien, Daniel Hueng-Yuan Shen, Li-Fan Lin, Chuang-Hsin Chiu, Cheng-Yi Cheng, Chien-An Sun, Ping-Ying Chang

**Affiliations:** 1Department of Nuclear Medicine, Tri-Service General Hospital and National Defense Medical Center, Taipei City 11490, Taiwan; x510107@hotmail.com (C.-H.K.); shen8484@gmail.com (D.H.-Y.S.); fanlin2@gmail.com (L.-F.L.); treasure316@gmail.com (C.-H.C.); chengcy60@gmail.com (C.-Y.C.); 2School of Public Health, National Defense Medical Center, Taipei City 11490, Taiwan; g694810042@gmail.com; 3Department of Medical Research, Tri-Service General Hospital and National Defense Medical Center, Taipei City 11490, Taiwan; 4School of Medicine, National Defense Medical Center, Taipei City 11490, Taiwan; 5Department of Public Health, College of Medicine, Fu-Jen Catholic University, New Taipei City 24205, Taiwan; 040866@mail.fju.edu.tw; 6Big Data Research Center, College of Medicine, Fu-Jen Catholic University, New Taipei City 24205, Taiwan; 7Division of Hematology/Oncology, Department of Internal Medicine, Tri-Service General Hospital and National Defense Medical Center, Taipei City 11490, Taiwan

**Keywords:** radioactive iodine, thyroid cancer, cardiovascular, comorbidity, mortality

## Abstract

(1) Background: This study aimed to investigate the association between radioactive iodine (RAI) and long-term cardiovascular disease (CVD) morbidity/mortality in thyroid cancer. (2) Methods: The study was conducted using data from the Taiwan National Health Insurance Database during 2000–2015. Thyroid cancer patients aged ≥20 years were categorized into RAI (thyroidectomy with RAI) and non-RAI (thyroidectomy only) groups. The Cox proportional hazard regression model and Kaplan–Meier method were used for analysis. (3) Results: A total of 13,310 patients were included. Kaplan–Meier analysis demonstrated that the two groups had similar cumulative risks of CVD (log-rank *p* = 0.72) and CVD-specific mortality (log-rank *p* = 0.62). On Cox regression analysis of different RAI doses, the risk of CVD was higher in the cumulative dosage >3.7 GBq (hazard ratio = 1.69, 95% confidence interval = 1.24–2.40, *p* < 0.001). (4) Conclusions: RAI was not associated with an increased risk of CVD in thyroid cancer. However, CVD surveillance is indicated in the patients receiving the cumulative RAI dosage above 3.7 GBq.

## 1. Introduction

Thyroid cancer is the fifth most common cancer in women in the USA, and an estimated 52,890 new cases are expected in 2020 [1]. In most countries, a steady increase in the incidence of thyroid cancer has been observed in both sexes [2]. Differentiated thyroid cancer, including papillary and follicular subtypes, is the most common variant and the standard treatment is surgery followed by indicated radioactive iodine (RAI) [3]. After treatment, thyroid hormone suppression therapy (THST) can reduce recurrence and cancer-related mortality [4].

RAI has some potential adverse effects, such as gastrointestinal symptoms, radiation thyroiditis, sialadenitis/xerostomia, bone marrow suppression, gonadal damage, dry eye, secondary cancers, pulmonary fibrosis, and genetic effects. Moreover, these adverse effects are associated with the cumulative RAI dosage [5]. Exposure of the arteries to radiation has been reported to cause or accelerate the development of atherosclerosis [6]. In addition to the known adverse effects, a study reported that the cumulative RAI dosage slightly increased the risk of atrial fibrillation (AF) [7]. Radiation has direct and indirect effects on our cells, and it is thought to be the mechanism of the known adverse effects [8]. However, it is not clear whether the RAI may directly damage the heart, blood vessels, or other organs and indirectly cause damage through reactive oxygen species produced by radiolysis of water. To our knowledge, the association between RAI and CVD is still not clear.

We conducted a longitudinal, nationwide, population-based, retrospective cohort study using the Taiwan National Health Insurance Research Database (NHIRD) to investigate whether RAI increases the subsequent risk of long-term cardiovascular morbidity and mortality in patients with thyroid cancer.

## 2. Materials and Methods

### 2.1. Data Sources

Data were extracted from the Longitudinal Health Insurance Database (LHID), a subset of the Taiwan NHIRD, which is a secondary database consisting of fully anonymous, comprehensive information, such as demographic data, dates of clinical visits, and disease diagnoses for 1 million enrollees, and is derived from the medical claims records of the National Health Insurance (NHI) program. The diagnostic codes in the NHIRD application forms are based on the International Classification of Diseases, Ninth Revision and Clinical Modification (ICD-9-CM). The Registry for Catastrophic Illness Patient Database (RCIPD) is also a subset of the NHIRD and includes data from insured residents with severe diseases defined by the NHI program, such as malignancies. The case group in this study comprised the patients included in the RCIPD. Under series review by the Institutional Review Board of Tri-Service General Hospital, the study was approved, and the requirement for informed consent was waived. The Approval No. of the study is TSGHIRB No.: B-109-19.

### 2.2. Study Design and Population

Between 1 January 2000 and 31 December 2015, we extracted data from the LHID and RCIPD for patients aged ≥20 years with complete information on age and sex, and with a history of thyroid cancer (ICD-9-CM 193). The patients with thyroid cancer who underwent total thyroidectomy (ICD-9-CM 06.4) followed by the THST (in compliance with the American Thyroid Association guidelines) without dose adjustment for more than one year were included. We defined the patient taking thyroid hormones without dosage adjustment above a year (we hypothesized that these patients could achieve their TSH level goal and were under stable thyroid hormone levels). Thyroid cancer patients were divided into two groups: the RAI group (total thyroidectomy with RAI (ICD-9-CM code 92.29)) and non-RAI group (total thyroidectomy only). We also analyzed the association between the different cumulative dosages of RAI (0.037–1.11 GBq/1–30 mCi, 1.147–3.7 GBq/31–100 mCi, 3.737–5.55 GBq/101–150 mCi, and >5.55 GBq/150 mCi) and the risk of CVD.

Study outcomes were CVD events and CVD-specific mortality, including ischemic heart diseases (ICD-9-CM 410–414), diseases of pulmonary circulation (ICD-9-CM 430–438), atrial fibrillation (AF, ICD-9-CM 427.31), other forms of heart diseases, excluding AF (ICD-9-CM 420–429), cerebrovascular diseases (ICD-9-CM 430–438), diseases of the arteries, arterioles, and capillaries (ICD-9-CM 440–449), diseases of the veins and lymphatics, and other diseases of the circulatory system (ICD-9-CM 451–459). The index date was defined as the time when thyroid cancer was diagnosed and the commencement of RAI treatment. We excluded patients with thyroid cancer before the index date, those who were diagnosed with CVD before the index date, other cancers (ICD-9-CM 140–208) before the index date, loss of tracking, age <20, and no specified gender.

### 2.3. Comorbidities

The following comorbidities were assessed: diabetes mellitus (DM, ICD-9-CM 250), chronic kidney disease (CKD, ICD-9-CM 585), hyperlipidemia (ICD-9-CM 272), and modified Charlson comorbidity index score (excluding myocardial injury, chronic heart failure, peripheral vascular diseases, cerebrovascular diseases or transient ischemic attack, and hemiplegia) [9].

### 2.4. Statistical Analysis

The Chi-square test was used for the analysis of categorical data. Continuous variables presented as the mean (± SD) were analyzed using the two-sample t-test. To investigate the risk of CVD, and CVD-specific mortality for patients with and without RAI treatment, a multivariate Cox proportional hazards regression analysis was used to calculate the hazard ratios (HRs) and 95% confidence intervals (CIs). The Kaplan–Meier method was used to determine the difference in the risk of CVD and CVD-specific mortality for the RAI group and non-RAI group using the log rank test. The variables were the comorbidities, such as gender, age, DM, CKD, hyperlipidemia, and modified Charlson comorbidity index score. All analyses were conducted using SPSS 22.0 software (SPSS Inc., Armonk, NY: IBM Corp.), with a two-tailed test *p* < 0.05 for statistical significance.

## 3. Results

### 3.1. Characteristics of the Control and Study Groups

We identified 19,016 thyroid cancer patients with total thyroidectomy. After applying the exclusion criteria, 5706 patients were excluded from the study. We divided the remaining 13,310 patients into RAI (11,889 patients) and non-RAI groups (1421 patients) (Figure 1). The baseline clinical characteristics between the two groups did not show significant differences except for CKD (*p* = 0.04, Table 1). The risks for patients with CVD and CVD-specific mortality at the end of study were comparable in RAI and non-RAI groups (11.8% vs. 11.5%, *p* = 0.40; 1.8% vs. 2.0%, *p* = 0.32; Appendix A). In the Kaplan–Meier analysis, RAI and non-RAI groups had similar cumulative risks of CVD (Log-rank *p* = 0.72, Figure 2a) and CVD-specific mortality (Log-rank *p* = 0.62, Figure 2b) (Appendix A). The mean tracking time was 5.16 years.

### 3.2. Risk Factors of CVD and CVD-Specific Mortality

In the Cox regression multivariate analysis of CVD and CVD-specific mortality, the risk of CVD and CVD-specific mortality was similar between RAI and non-RAI groups (CVD: HR = 0.99, 95% CI = 0.84–1.16, *p* = 0.88; CVD-specific mortality: HR = 0.92, 95% CI = 0.62–1.37, *p* = 0.68). Males had a higher risk of CVD than females (HR = 1.32, 95% CI = 1.17–1.48, *p* < 0.001). Meanwhile, age, CKD, and hyperlipidemia were independent risk factors for both CVD and CVD-specific mortality (all: *p* < 0.001, Table 2, Appendix A).

### 3.3. Cumulative RAI Dosages and Risks of CVD and CVD-Specific Mortality

The effect of cumulative RAI dosage is summarized in Table 3. The risk of CVD was significantly higher in patients with the cumulative dosage between 3.737 and 5.55 GBq and greater than 5.55 GBq (HR = 1.56, 95% CI = 1.17–2.12, *p* < 0.001; HR = 1.69, 95% CI = 1.24–2.40, *p* < 0.001, respectively). Patients with the cumulative dosage greater than 5.55 GBq had a significantly higher risk of CVD-specific mortality (HR = 1.75, 95% CI = 1.10–2.28, *p* = 0.001).

### 3.4. Risks of Subgroups of CVD and CVD-Specific Mortality

The RAI group had similar risks of developing CVD and CVD-specific mortalities from any subgroup compared to that of the non-RAI group (Table 4). The RAI group also did not display a higher risk of developing AF (HR = 1.11, 95% CI = 0.94–1.31, *p* = 0.14) and AF-related mortality (HR = 0.92, 95% CI = 0.78–1.08, *p* = 0.80).


## 4. Discussion

Our study demonstrated that RAI treatment was not associated with a higher risk of CVD or CVD-specific mortality compared to non-RAI cases, including that of specific disease subgroups. We found that the cumulative dosage more than 3.7 GBq was associated with a higher risk of developing CVD, and the cumulative dosage more than 5.55 GBq was associated with a higher risk of CVD-specific mortalities.

Increased CVD morbidity and mortality have been reported among well-differentiated thyroid carcinoma patients, and THST was considered as a major cause of the cardiovascular problems [7,10,11,12,13,14,15,16,17,18,19,20,21]. Klein Hesselink et al. reported that a lower TSH level is associated with increased CVD mortality in thyroid cancer patients [10]. However, a marginally increased risk of AF (sub-distribution hazard ratio: 1.04, 95% CI: 1.01–1.08, *p* = 0.006) was associated with the cumulative dosage of RAI independent of the TSH level [7]. Abonowara et al. found an increased prevalence of AF among thyroid cancer patients, but found no correlation between the TSH level and the occurrence of AF [11]. Wang et al. showed that there was no difference in the risk of AF among thyroid cancer patients with THST [12]. The optimal intensity and duration of THST has not been confirmed. Moreover, THST-related subclinical hyperthyroidism may have adverse effects on the bones and the heart [13]. In our study, RAI treatment did not increase the risk of CVD morbidity and mortality. In addition, the risk of AF was comparable in the RAI and non-RAI groups (HR = 1.11, 95% CI = 0.94–1.31, *p* = 0.14). The American Thyroid Association (ATA) 2015 guideline recommends maintaining TSH below 0.1 mU/L in thyroid cancer patients with structural incomplete response and keeping TSH within the low reference range (0.5–2 mU/L) in those with excellent response or indeterminate response with low risk of recurrence [22]. Earlier editions of the ATA guidelines recommended that TSH be maintained at a lower level (0.1–0.5 mU/L), which may explain the occurrence of increased CVD adverse events observed in previous studies.

The study conducted by la Cour et al. demonstrated that there was an increased risk of cerebrovascular problems in both hyperthyroid and euthyroid patients treated with RAI for benign thyroid disorders [6]. They proposed that RAI treatment may expose the carotid arteries to radiation, which could initiate or accelerate atherosclerosis. In their subsequent study, which evaluated changes in carotid intima thickness and plaque formation after RAI, they failed to demonstrate the increase in atherosclerotic burden on the carotid arteries from radiation exposure [23]. This was explored by Sanal et al., who reported that RAI may increase the carotid intima media thickness in thyrotoxicosis patients [24]. The relationship between RAI, carotid intima thickness, and subsequent atherosclerosis in thyroid cancer patients is still controversial. In our study, RAI was not associated with a higher risk of cerebrovascular disease (HR = 1.04, 95% CI = 0.74–1.20, *p* = 0.91), which is consistent with the established literature [25].

Our study showed RAI cumulative dosage more than 3.7 GBq was associated with higher risk of CVD comorbidity and the cumulative dosage more than 5.55 GBq was associated with higher risk of CVD-related mortality. RAI is taken up by cells with a membrane sodium–iodide transporter, Na^+^/I^−^ symporter (NIS). NIS-mediated I^−^ uptake plays a pivotal role in thyroid hormone biosynthesis in thyroid cells [26,27]. NIS gene expression was also found in some extrathyroidal tissues, including in the heart [28]. The uptake of circulating RAI in cardiac tissue may lead to the injuries observed. In experimental models, radiation doses ≥2 Gy will induce the expression of inflammatory cytokines and adhesion molecules in the endothelial cells of the arteries and initiate the process of atherosclerosis [29,30]. Radiation doses ≥8 Gy will increase the number and size of the atherosclerotic plaques that are prone to rupture [30,31]. Therefore, the cumulative dosage of RAI may have deleterious effects on the cardiovascular system. In addition, RAI treatment is not routinely suggested for patients with ATA low-risk disease [22]. These patients may take relatively lower dosages of levothyroxine to achieve the TSH goal. On the other hand, the patients with recurrent thyroid cancer may not only have a higher cumulative dosage of RAI but also more periods of thyroid hormone withdrawal, with concomitant periods of hypothyroidism. An et al. reported that thyroid hormone withdrawal-related hypothyroidism may induce pronounced changes in various cardiometabolic parameters and increase the risk of atherosclerosis and cardiovascular diseases [32].

This study has several limitations. First, we enrolled the patients from the NHIRD with diagnostic code ICD-9-CM. The clinicopathologic laboratory parameters (especially the data of serum TSH level and serum thyroid hormones level), and detailed personal information regarding factors such as smoking habits, body mass index, and family history, were not available. To minimize the bias of a large population-based database, levothyroxine dosage has been used as a surrogate marker of TSH suppression [18]. In our study, if the dosage of levothyroxine remained unchanged for more than one year, we hypothesized that these patients might achieve their own goals of TSH level and be under stable thyroid function without clinical adverse effects. Second, we could not classify the patients according to individual prognostic factors, including the American Joint Committee on Cancer staging system and the ATA risk stratification system. However, above 95% of thyroid cancer cases in Taiwan are diagnosed as differentiated thyroid cancer at the localized stage [33,34,35]. Our study matched the gender and age in the study (RAI) and control (non-RAI) groups and focused on the impact of RAI cumulative dose on the risk of long-term CVD morbidity and mortality. It is less likely that the thyroid cancer type and/or stage affect the risk of CVDs. Finally, although the cumulative dosage of RAI was determined, we did not know the dose density of RAI, which may cause different biological effects. Despite these limitations, the strength of the study is that we report longitudinal results of the association between RAI treatment and the risk of cardiovascular morbidity and mortality in a nationwide population-based cohort.

## 5. Conclusions

This study showed that RAI treatment was not associated with an increased CVD risk in thyroid cancer patients. However, our results stress the importance of a close follow-up and encourage assessment and treatment of CVD risk factors during follow-up, especially for the cumulative dose more than 3.7 GBq.

## Figures and Tables

**Figure 1 jcm-10-04032-f001:**
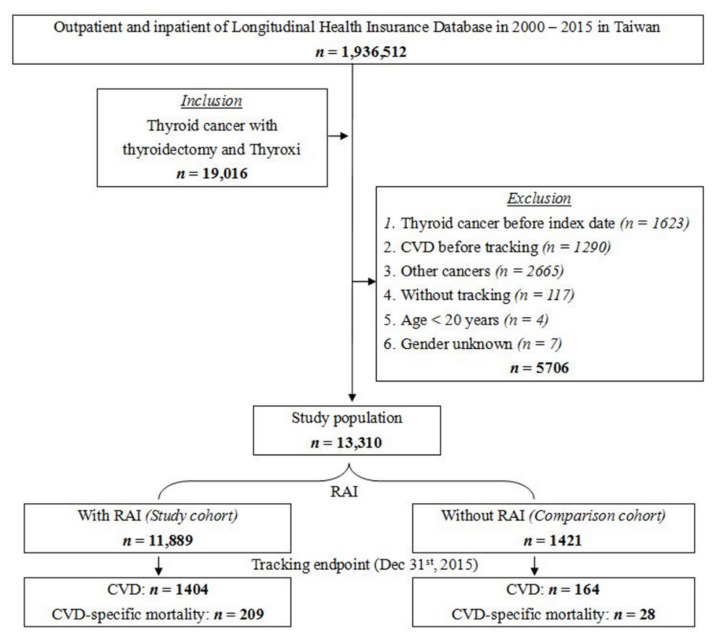
Flowchart of the study design.

**Figure 2 jcm-10-04032-f002:**
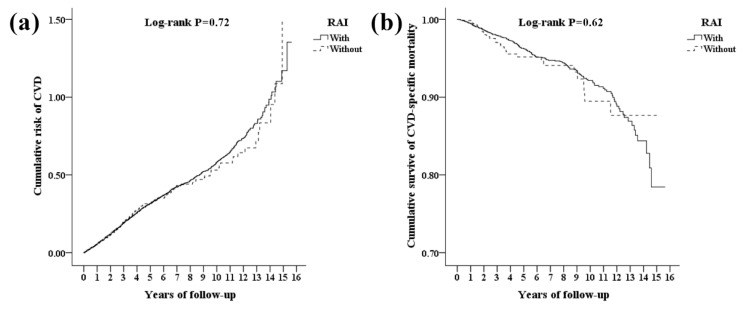
Kaplan–Meier plots for the cumulative risk of cardiovascular diseases (CVD) (**a**) and cumulative survival of CVD-specific mortality stratified by radioiodine (RAI) with log-rank test (**b**).

**Table 1 jcm-10-04032-t001:** Characteristics of the control and study groups at baseline.

**Variables**		**RAI Group**		**Non-RAI Group**		***p***
		***n***	**%**	***n***	**%**	
	**Total**	11,889	89.3	1421	10.7	
**Gender**	Male	2259	19.0	289	20.3	0.12
	Female	9630	81.0	1132	79.7	
**Age (years)**		47.3 ± 13.2		47.2± 13.2		0.70
**Age groups (years)**	20–39	3673	30.9	437	30.8	1.00
	40–59	6217	52.3	744	52.4	
	60–79	1910	16.1	229	16.1	
	≥80	89	0.8	11	0.8	
**DM**	Without	11,216	94.3	1350	95.0	0.17
	With	673	5.7	71	5.0	
**CKD**	Without	11,757	98.89	1413	99.4	0.04
	With	132	1.1	8	0.6	
**Hyperlipidemia**	Without	11,752	98.85	1408	99.1	0.25
	With	137	1.2	13	0.9	
**Duration of levothyroxine without dosage adjustment (years)**	1 ≤ years <2	1815	15.3	219	15.4	0.11
	2 ≤ years < 6	4249	35.7	552	38.9	
	6 ≤ years <11	3328	28.0	364	25.6	
	≥11 years	2497	21.0	286	20.1	
**CCI_R**		0.08 ± 0.28		0.07 ± 0.28		0.25

P: Chi-square/Fisher exact test on category variables and t-test on continued variables. RAI = radioactive iodine; DM = diabetes mellitus; CKD = chronic kidney disease; CCI_R = Charlson comorbidity index after removal of the above-mentioned comorbidities and myocardial injury, chronic heart failure, peripheral vascular diseases, cerebrovascular diseases or transient ischemic attack, and hemiplegia; SD = standard deviation.

**Table 2 jcm-10-04032-t002:** Factors of CVD and CVD-specific mortality by using Cox regression.

Outcome		CVD			CVD-Specific Mortality		
Variables		Adjusted HR	95% CI	*P*	Adjusted HR	95% CI	*p*
**RAI**	Without	Reference			Reference		
	With	0.99	0.84–1.16	0.88	0.92	0.62–1.37	0.68
**Gender**	Female	Reference			Reference		
	Male	1.32	1.17–1.48	<0.001	1.22	0.91–1.63	0.19
**Age groups (yrs)**	20–39	Reference			Reference		
	40–59	1.66	1.37–2.00	<0.001	2.58	1.09–6.13	0.03
	60–79	2.73	0.26–0.30	<0.001	10.42	4.55–23.87	<0.001
	≥80	4.42	3.56–5.48	<0.001	31.48	13.62–72.75	<0.001
**DM**	Without	Reference			Reference		
	With	0.84	0.70–1.01	0.06	0.63	0.39–1.04	0.07
**CKD**	Without	Reference			Reference		
	With	1.85	1.51–2.27	<0.001	3.94	2.75–5.64	<0.001
**Hyper-** **lipidemia**	Without	Reference			Reference		
	With	1.538	1.24–1.91	<0.001	0.12	0.02–0.83	0.03
**CCI_R**		1.43	1.28–1.60	<0.001	0.98	0.71–1.34	0.88

P: Chi-square/Fisher exact test on category variables and t-test on continued variables. RAI = radioactive iodine; CVD: cardiovascular disease; DM = diabetes mellitus; CKD = chronic kidney disease; CCI_R = Charlson comorbidity index after removal of the above-mentioned comorbidities and myocardial injury, chronic heart failure, peripheral vascular diseases, cerebrovascular diseases or transient ischemic attack, and hemiplegia; CI, confidence interval; HR: hazard ratio.

**Table 3 jcm-10-04032-t003:** Factors of CVD and CVD-specific mortality among different dosages of RAI using Cox regression.

Outcome	Dose of RAI	Populations	Events	PYs	Rate	Adjusted HR	95% CI	*p*
**CVD**	Without	1421	164	2744.86	5974.81	Reference		
	With	11,889	1404	22,899.15	6131.23	0.99	0.84–1.16	0.88
	0.037–1.11 GBq	5491	503	9678.65	5197.00	0.97	0.59–1.03	0.85
	1.147–3.7 GBq	678	155	2468.54	6279.01	1.30	0.94–1.74	0.73
	3.737–5.55 GBq	231	99	1482.99	6675.72	1.56	1.17–2.12	<0.001
	>5.55 GBq	5489	647	9268.97	6980.28	1.69	1.24–2.40	<0.001
**CVD-** **specificmortality**	Without	1421	28	3058.93	915.35	Reference		
	With	11,889	209	25,642.17	815.06	0.92	0.62–1.37	0.68
	0.037–1.11 GBq	5491	72	10,989.12	655.19	0.75	0.39–1.19	0.70
	1.147–3.7 GBq	678	20	2601.86	768.68	0.88	0.45–1.36	0.51
	3.737–5.55 GBq	231	17	1768.25	961.40	1.28	0.80–1.78	0.29
	>5.55 GBq	5489	100	10,282.94	972.48	1.75	1.10–2.28	0.001

^3^ RAI = radioactive iodine; CVD = cardiovascular disease; PY = person-years; Rate = per 100,000 person-years; CI = confidence interval; HR = hazard ratio.

**Table 4 jcm-10-04032-t004:** Factors of CVD and CVD-specific mortality subgroup stratified by variables listed in the table by using Cox regression.

	**RAI Group**			**Non-RAI Group**			**RAI vs. Non-RAI** *(Reference)*			
Outcome Subgroup	Events	PYs	Rate	Events	PYs	Rate	Ratio	Adjusted HR	95% CI	*P*
**Any combination of the listed CVD**	1404	22,899.15	6131.23	164	2744.86	5974.81	1.03	0.99	0.84–1.16	0.88
Ischemic heart diseases	570	24,463.32	2330.02	63	2933.78	2147.40	1.09	1.02	0.78–1.32	0.89
Diseases of pulmonary circulation	59	25,529.81	231.10	6	3054.33	196.44	1.18	1.13	0.49–2.64	0.77
AF	68	24,602.99	276.39	7	2910.87	240.48	1.15	1.11	0.94–1.31	0.14
Other forms of heart disease	497	24,607.98	2019.67	60	2876.80	2085.65	0.97	0.93	0.79–1.10	0.73
Cerebrovascular diseases	385	24,857.84	1548.81	43	2998.36	1434.12	1.08	1.02	0.74–1.40	0.91
Diseases of arteries, arterioles, and capillaries	70	25,497.55	274.54	7	3048.98	229.59	1.20	1.17	0.53–2.55	0.70
Veins and lymphatics, and other	337	25,024.94	1346.66	38	2997.21	1267.84	1.06	1.08	0.77–1.51	0.66
CVD-specific mortality	209	25,642.17	815.06	28	3058.93	915.35	0.89	0.92	0.62–1.37	0.68
Ischemic heart diseases	55	25,642.17	214.49	10	3058.93	326.91	0.66	0.70	0.35–1.39	0.30
Diseases of pulmonary circulation	10	25,642.17	39.00	0	3058.93	0.00	∞	∞	-	0.84
AF	10	25,642.17	39.00	1	3058.93	32.69	1.193	1.15	0.97–1.35	0.08
Other forms of heart diseases	88	25,642.17	343.18	11	3058.93	359.60	0.954	0.92	0.78–1.08	0.80
Cerebrovascular diseases	44	25,642.17	171.59	6	3058.93	196.15	0.88	0.84	0.35–2.00	0.70
Diseases of arteries, arterioles, and capillaries	9	25,642.17	35.10	1	3058.93	32.69	1.07	1.09	0.14–8.80	0.94
Veins and lymphatics, and other	32	25,642.17	124.79	3	3058.93	98.07	1.27	1.18	0.36–3.90	0.78

RAI = radioactive iodine; CVD = cardiovascular disease; AF = atrial fibrillation; PY = person-years; Rate = per 1000 person-years; CI = confidence interval; HR = hazard ratio.

## Data Availability

The study extracted data from the Taiwan National Health Insurance Research Database.

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
