# Peer review of "Radioactive Iodine Treatment and the Risk of Long-Term Cardiovascular Morbidity and Mortality in Thyroid Cancer Patients: A Nationwide Cohort Study"

_jcm, 2021, doi:10.3390/jcm10174032_

Round 1
Reviewer 1 Report
A well written manuscript with clear messages. Attached some comments:
- Line 78: thyroid hormone suppression therapy - abbreviation already defined above (line 41)
- Line 85-86: how were cumulative dosages of RAI calculated? Did LHID and RCIPD databases offer data on this?
- Line 93-95: shouldn't the index date be the same between RAI and nonRAI groups? E.g., the date of cancer diagnosis for both groups...
- Line 108: Please state in methods section which variables were included in the multivariable models.
- Line 119-120: Why did RAI and nonRAI groups differ regarding CKD? Is CKD a contraindication for RAI?
- In Table 4 first column "any of the listed" and "respectively" are stated. Could you make more clear what you mean? It takes some time to understand...
- Was the follow-up enough (5 years according to suppl table 2) to reveal differences in CVD morbidity and mortality?
- Line 178-179 THST was considered as a major cause of the cardiovascular problems [7,10-21].: could you offer a sentence describing the hypothetized mechanism?
- Line 201 correct plague to plaque
Author Response
Response to Reviewer 1 Comments
Dear reviewers,
Thank you for your precious comment to our manuscript. The revised manuscript is uploaded as other file, and there are the responses to the questions:
Point 1: Line 78: thyroid hormone suppression therapy - abbreviation already defined above (line 41). 

Response 1: We corrected this in line 78 of the revised manuscript.
Point 2: Line 85-86: how were cumulative dosages of RAI calculated? Did LHID and RCIPD databases offer data on this?
Response 2: Yes. LHID is a claim database. All the treatments, including the dose of RAI, should be declared in the system of National Health Insurance in Taiwan. The cumulative dosages of RAI are available in the LHID.
Point 3: Line 93-95: shouldn't the index date be the same between RAI and nonRAI groups? E.g., the date of cancer diagnosis for both groups...
Response 3: Thank you for the important question. In generally, RAI treatment may be performed about 1-2 months after the total thyroidectomy operation, according to the recovery status of the patient, previous iodine-based contrast materials injection date (for computed tomography study) and thyroxine withdrawal duration for the TSH stimulation. Besides, the actual event in RAI group is the date of first treatment of RAI. Thus, we defined the index date as the time when thyroid cancer was diagnosed and the commencement of RAI treatment.
Point 4: Line 108: Please state in methods section which variables were included in the multivariable models.
Response 4: The variables were the comorbidities such as gender, age, DM, CKD, hyperlipidemia and modified Charlson comorbidity index score (exclude DM, CKD and myocardial injury, chronic heart failure, peripheral vascular diseases, cerebrovascular diseases or transient ischemic attack and hemiplegia). We added in line 110-112 of the revised manuscript.
Point 5: Line 119-120: Why did RAI and nonRAI groups differ regarding CKD? Is CKD a contraindication for RAI?
Response 5:
- RAI is predominantly cleared by renal excretion, and the clearance is reduced in the patients with CKD, However, RAI is not contraindicated in CKD [1].
- There were few events of CKD in the two groups (1.1% in the RAI and 0.6% in the nonRAI). Though it showed significant differences (P=0.04, <0.05), the significance remained uncertain.
Point 6: In Table 4 first column "any of the listed" and "respectively" are stated. Could you make more clear what you mean? It takes some time to understand...
Response 6: Thank you for the comments. A case might be diagnosed with 2 diseases in the same outcome subgroup [e.g. acute myocardial infarction (ICD-9-CM 410) and coronary atherosclerosis (ICD-9-CM 414)]. To make this easier to understand, we changed “any of the listed” to “any combination of the listed”. Please see the revised table 4.
Point 7: Was the follow-up enough (5 years according to suppl table 2) to reveal differences in CVD morbidity and mortality?
Response 7: Thank you for the important question. In the study, we extracted data from the LHID and RCIPD for patients between January 1, 2000, and December 31, 2015. It is reasonable that some cases may not reach the end point of the study due to the event (CVD-specific mortality). We replaced the “median years” with the “Detail numbers in each year of Kaplan-Meier plots for the cumulative risk of CVD and cumulative survival of CVD-specific mortality stratified by RAI with log-rank test.” as the revised suppl table 2. There is no significant difference in each tracking time. We thought it would be more clearly and the follow-up duration in the study was enough to reveal the result.
Point 8: Line 178-179 THST was considered as a major cause of the cardiovascular problems [7,10-21].: could you offer a sentence describing the hypothetized mechanism?
Response 8: Our hypothesis “Moreover, THST related subclinical hyperthyroidism may have adverse effects on the bones and the heart” was in line 188-189 of the revised manuscript.
Point 9: Line 201 correct plague to plaque
Response 9: We corrected this in line 202 of the revised manuscript.
Thank you for your precious advice, which help us to improve the shortcomings in the manuscript.
Sincerely
Reference
- Haugen, B.R.; Alexander, E.K.; Bible, K.C.; Doherty, G.M.; Mandel, S.J.; Nikiforov, Y.E.; Pacini, F.; Randolph, G.W.; Sawka, A.M.; Schlumberger, M.; et al. 2015 American Thyroid Association Management Guidelines for Adult Patients with Thyroid Nodules and Differentiated Thyroid Cancer: The American Thyroid Association Guidelines Task Force on Thyroid Nodules and Differentiated Thyroid Cancer. Thyroid 2016, 26, 1-133, doi:10.1089/thy.2015.0020.

Reviewer 2 Report
In this article the question in terms of public health posed is relevant and of great importance. Indeed, this epidemiological study deals with a health question of importance in medicine.
This study is complementary to other studies published in the literature in recent years on the long-term effects of the treatment or use of radioactive iodine in nuclear medicine.
Highlights:
-the protocol is correct, the cohorts are correct, the analytical strategy is well constructed.
-in this study, scientific and medical information is a plus for clnicians-practitioners.
-this national study (Taiwan) is reassuring for the treatment of patients with thyroid cancer with radioactive iodine.
-although there is no association between radioactive iodine treatment and long-term conditions, this study makes recommendations on the follow-up and risk assessment of patients treated.
Author Response
Response to Reviewer 2 Comments
Dear reviewers,
Thank you for your precious comment to our manuscript. We minorly revised our manuscript according to other reviewers’ comments and is uploaded as other file. No major revision was done. Thank you for your help to improve the shortcomings in the manuscript.
Sincerely
